# A micro-epidemiological analysis of febrile malaria in Coastal Kenya showing hotspots within hotspots

Philip Bejon[1,2]*, Thomas N Williams[1,3], Christopher Nyundo[1], Simon I Hay[4], David Benz[4], Peter W Gething[4], Mark Otiende[1], Judy Peshu[1], Mahfudh Bashraheil[1], Bryan Greenhouse[5], Teun Bousema[6,7], Evasius Bauni[1], Kevin Marsh[1,2], David L Smith[8], Steffen Borrmann[1,9,10]

[1]KEMRI-Wellcome Trust Research Programme, Kilifi, Kenya; [2]Centre for Clinical Vaccinology and Tropical Medicine, University of Oxford, Oxford, United Kingdom; [3]Imperial College London, London, United Kingdom; [4]Spatial Ecology and Epidemiology Group, Department of Zoology, University of Oxford, Oxford, United Kingdom; [5]Department of Medicine, University of California, San Francisco, San Francisco, United States; [6]Department of Medical Microbiology, Radboud University Nijmegen Medical Centre, Nijmegen, Netherlands; [7]London School of Hygiene and Tropical Medicine, London, United Kingdom; [8]John Hopkins Malaria Research Institute, Baltimore, United States; [9]Institute for Tropical Medicine, University of Tübingen, Germany; [10]German Centre for Infection Research, Tübingen, Germany

*For correspondence: pbejon@kemri-wellcome.org

Competing interests: The authors declare that no competing interests exist.

**Abstract** Malaria transmission is spatially heterogeneous. This reduces the efficacy of control strategies, but focusing control strategies on clusters or 'hotspots' of transmission may be highly effective. Among 1500 homesteads in coastal Kenya we calculated (a) the fraction of febrile children with positive malaria smears per homestead, and (b) the mean age of children with malaria per homestead. These two measures were inversely correlated, indicating that children in homesteads at higher transmission acquire immunity more rapidly. This inverse correlation increased gradually with increasing spatial scale of analysis, and hotspots of febrile malaria were identified at every scale. We found hotspots within hotspots, down to the level of an individual homestead. Febrile malaria hotspots were temporally unstable, but 4 km radius hotspots could be targeted for 1 month following 1 month periods of surveillance.

## Introduction

The transmission of infectious disease often shows substantial heterogeneity (*Woolhouse et al., 1997*). Malaria transmission is determined by mosquito ecology and behavior, which is in turn determined by rainfall, hydrology, soils, human behavior and population distributions, and a range of other social, biotic and abiotic factors. Heterogeneity of malaria transmission is apparent at global scale (*Gething et al., 2011*), regional scale (*Kleinschmidt et al., 2001a*; *Noor et al., 2009*), and at fine scale in, for instance, Mali (*Gaudart et al., 2006*), Ghana (*Kreuels et al., 2008*), Ethiopia (*Yeshiwondim et al., 2009*) Kenya (*Brooker et al., 2004*; *Ernst et al., 2006*; *Bejon et al., 2010*), and Tanzania (*Bousema et al., 2010*). This spatial heterogeneity makes transmission relatively resilient to indiscriminate control efforts, but also provides an opportunity to engage in targeted malaria control on clusters of transmission (or 'hotspots'), a strategy that is predicted to be highly effective (*Dye and Hasibeder 1986*; *Woolhouse et al., 1997*).

**eLife digest** Malaria remains a formidable threat to public health in tropical regions. The parasite that causes the disease is transmitted to humans by bites from infected mosquitoes, and the complicated lifecycle of the parasite makes developing vaccines difficult. However, preventive strategies are effective at reducing the spread of malaria. The two most widely used and effective strategies are the use of pesticide-treated bed nets to create a barrier between sleeping families and biting mosquitoes, and indoor residual spraying to reduce the numbers of mosquitoes biting sleeping families in homesteads. Other potential preventive strategies include killing mosquito larvae in breeding sites and mass anti-malarial drug treatment for infected humans.

Targeting preventive efforts to malaria hotspots—the areas where the risk of malaria transmission is greatest—may help to eliminate malaria more efficiently. Unfortunately, identifying hotspots is complicated as there are many different factors that affect how malaria spreads. These factors range from ecological conditions such as rainfall and soil type, to human effects like population density and migration.

Bejon et al. have examined the patterns of malaria transmission in Kenya over 9 years. Over this period, 54% of children who went to health clinics with a fever tested positive for the parasite that causes malaria. Infected children from areas with the highest rate of malaria infection were, on average, younger than those from less infected regions. This makes sense as in highly affected areas children have a greater chance of encountering the parasite at an early age. They are therefore more likely to get malaria when younger and, as exposure to the parasite can provide some immunity to a child, they are also less likely to get infected again when older.

In addition, mapping the spread of malaria reveals hotspots at different geographical scales. Bejon et al. could see hotspots within hotspots, and in some cases could go as far as identifying the individual homesteads most at risk of malaria. Public health workers could potentially use these analyses to identify areas that are likely to be hotspots and then target preventive measures there for the next month. However, the constantly changing locations of the hotspots means workers would have to reanalyse the data and retarget their interventions at the end of each month.

We have previously identified hotspots of malaria using active surveillance (*Bejon et al., 2010*). Others have identified hotspots using passive surveillance in health facilities linked to demographic surveillance systems (*Ernst et al., 2006*). Passive surveillance is more readily scaled up, but may be biased by variations in access to health care facilities and socially-determined health-seeking behavior (*Sumba et al., 2008*; *Franckel and Lalou 2009*). The incidence of febrile malaria presenting to health care is thus biased by access to care. This bias may be countered by using the malaria positive fraction (MPF) among children with fever (also termed 'slide positivity rate' in some publications [*Jensen et al., 2009*]). The MPF includes all febrile children presenting to the dispensary as the denominator, hence controlling for access to health care, in contrast to incidence for which all children in the community are included in the denominator. The MPF is less likely to show systematic spatial bias with distance from the health facility since parental accounts of illness have not been found to discriminate malaria from non-malarial fever (*Luxemburger et al., 1998*; *Mwangi et al., 2005*), and diagnostic testing is not available outside the dispensary.

We present data from demographic surveillance linked to passive case detection in Pingilikani dispensary in Kilifi District, coastal Kenya. Data are collected from 1500 homesteads within an 8 km radius followed for 9 years. We analyse the spatial heterogeneity of malaria cases in order to determine the temporal and spatial scales of case clustering so as to inform targeting in malaria control programmes. We also excluded visits with specific symptoms such as skin infections or cutaneous abscesses, otitis media, and gastroenteritis (>4 episodes diarrhoea per day) that might have been the primary motivation for seeking health care rather than fever per se.

## Results

Among ~20,000 remaining febrile presentations from ~1500 different residences, 54% were positive for *Plasmodium falciparum* on blood smear examination. Using homestead as our unit of analysis, we found that the incidence of dispensary attendance declined with distance from the dispensary

(on average −0.040 (95%CI 0.036–0.044) and −0.041 (95%CI 0.037–0.046) episodes per child year for each km for malaria smear positive and negative attendees, respectively). MPF was not found to vary significantly by distance of residence from the dispensary (from MPF = 0.50, 95%CI 0.47 to 0.54 at <2 km distance to MPF = 0.52, 95%CI 0.47 to 0.57 at 6–7 km, p=0.7).

The spatio-temporal distribution of MPF by homestead is shown in *Video 1* (slow speed) and *Video 2* (fast speed). The visual impression from these clips suggests marked spatial variation, with some geographical areas showing persistently high MPFs, and other areas showing more marked temporal variation. Temporally stable spatial heterogeneity would be expected to lead to spatial heterogeneity in the acquisition of immunity, which may be evidenced by variation in the age profiles of children with febrile malaria. We therefore tested this hypothesis as below.

## Spatial heterogeneity in malaria risk and acquisition of immunity

MPF was inversely correlated with the average age of children with malaria, Spearman's rank correlation ($r_s$) = −0.16, p<0.0001 (*Figure 1A–C*). This suggests that greater exposure to malaria (i.e., high MPF) leads to more rapid acquisition of immunity as children grow up, hence predominantly younger children visiting the dispensary with febrile malaria. There was no evidence that this relationship was confounded by spatial clustering of age: the average age of children with non-malarial fever did not show spatial clustering (Moran's I = 0.01, p=0.5 within 1 km and Moran's I = 0.02, p=0.5 within 5 km) and was not associated with MPF ($r_s$ = −0.02, p=0.4). We examined the effect of spatial scale at which this correlation occurred by imposing grids of increasing cell size on the study area, calculating $r_s$ within each cell of the grid, and then estimating the mean $r_s$ at each scale of grid (*Figure 1D*, blue lines). The mean $r_s$ trended gradually away from 0 as the grid divisions became larger in scale. This pattern suggests gradual differentiation in transmission characteristics as the distance between homesteads included within a cell of the grid increases. We then examined the patterns seen on applying this analysis to simulated data. In order to exclude that this trend was a result of cells at fine-scale containing fewer homesteads, we ran permutations of the data using after randomly re-assigning spatial coordinates to the homesteads. These permutations show that a consistent correlation at $r_s$ = −0.16 throughout the range of grid sizes, albeit with greater uncertainty with smaller cell size (*Figure 1D*, red lines). Hence, the trend of a gradually increasing inverse correlation as the grid size increases does not appear to be explained simply by having fewer homesteads in each cell at fine scale. In order to determine the pattern that might be seen with specific spatial scales of clustering, we conducted further simulations by imposed patterns with specific scales on the spatial coordinates of the homesteads, in varying proportions with random noise using a gamma distribution. These simulations show that a specific scale of clustering produces 'spikes' in $r_s$ as the cell size varies, with the position of the spike coinciding with scale of the clustering (*Figure 1—figure supplement 1*). Reducing the Signal:Noise ratio eventually obscured the 'spikes' due to a characteristic pattern, but only at the point where the overall correlation was no longer discernible (*Figure 1—figure supplement 2*). Adding a gradient to the simulated characteristic scale attenuated but did not obscure the 'spikes' (*Figure 1—figure supplement 3*).

## Hotspots within hotspots

Using the Bernoulli model in SaTScan (*Kulldorff, 1997*), we identified a hotspot with a radius of 5.8 km at p<0.00001 (*Figure 2A*) using the full data set (for which n = 20,702). However, on re-analysis of the children within this hotspot (in which n = 5300), we identified a further hotspot (with a radius of 0.76 km) within the 5.8 km hotspot (p<0.00001, *Figure 2B*). Then on further re-analysis of the homesteads within that 0.76 km hotspot (within which n = 1406), we identified a third significant hotspot (p=0.016) which comprised a single homestead, in which there were 36 episodes of malaria compared with 3 malaria negative fevers (*Figure 2D*). When we selected a random 5-km square area

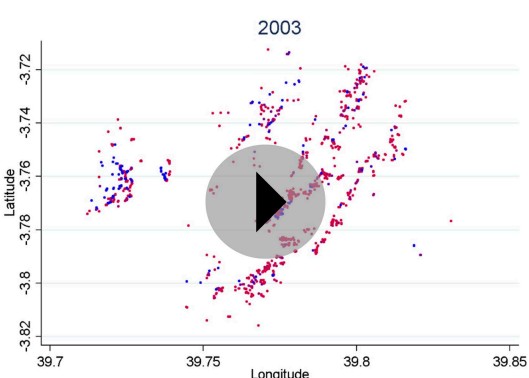

**Video 1**. Each plotted point represents an individual homestead, where the colour shading indicates the malaria positive fraction (MPF), with red shading for high MPF and blue shading for low MPF. Points change colour each year.

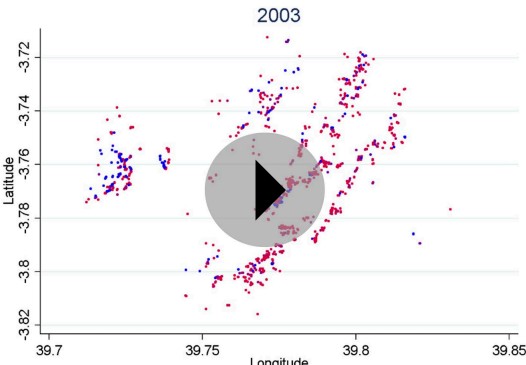

**Video 2**. Each plotted point represents an individual homestead, where the colour shading indicates the malaria positive fraction (MPF), with red shading for high MPF and blue shading for low MPF. Points change color each year. The frames are identical to those in **Video 1**, but move more rapidly.

outside the original 5.8 km radius hotspot, we identified a hotspot within this area a fourth hotspot with a 1.32 km radius (p<0.00001, **Figure 2C**).

To further explore the scale of spatial clustering, we plotted the semivariogram (**Figure 2—figure supplement 1**) and the log–log transformed semivariogram (**Figure 2—figure supplement 2**). These plots suggested linear fits for the semivariogram, suggesting that spatial clustering occurred over a range of spatial scales.

## Temporal trends of spatial heterogeneity

We also examined temporal trends for individual homesteads (**Figure 3**). There was an inverse correlation between the mean MPF and the variance in MPF over the 10-year study period ($r_s$ = −0.61, p<0.0001, **Figure 3A**). The temporal trends for two subsets of homestead can be seen in **Figure 3B** (stable high MPF) and **Figure 3C** (unstable low MPF), suggesting that homesteads can be characterized as stable high transmission homesteads or unstable low transmission homesteads. Infant parasite rates have been proposed as a measure of transmission intensity that minimizes the offsetting of acquired immunity in macro-epidemiological studies (**Snow et al., 1996**). We therefore hypothesized that the malaria positive fractions in children <1 year of age (hereafter 'MPF$_{<1yr}$') would measure transmission intensity without the offsetting of acquired immunity, and that unstable transmission would result in higher risk of malaria in older children. To test this hypothesis, we calculated the mean MPF$_{<1yr}$ and the variance in MPF$_{<1yr}$ for each homestead over the 9 years of follow up and tested the relationships between these metrics and risk of malaria in older children in multivariable linear regression models.

In multivariable linear regression models, MPF$_{<1yr}$ was strongly correlated with MPFs in children in the 1- to 2-year-old and 2- to 3-year-old age group, but progressively less strongly correlated with MPF in older children (**Figure 3Di**). The regression coefficient was ~0.4 for 1–2 year olds, meaning that each unit increase in MPF$_{<1yr}$ is associated with a 0.4 increase in the MPF for 1- to 2-year-old children. On the other hand, the variance in MPF$_{<1yr}$ was not correlated with MPFs in 1- to 2- or 2- to 3-year-old children, but was progressively more strongly correlated with MPF in older children (**Figure 3Dii**). Hence there were high stable transmission homesteads, with predominantly younger children getting febrile malaria, and low unstable transmission homesteads, with increasing risk to older children. This pattern of high stable vs low unstable transmission also occurs between regions or countries, and demonstrates a similarity between the micro- and macro-epidemiology of malaria (**Hay et al., 2008**).

## Theoretical accuracy of targeted control undertaken at varying temporal and spatial scales

We then used our data set to simulate the accuracy of targeting cases that a malaria control programme might achieve on conducting surveillance over a defined period of time followed by targeted control. We assumed that malaria control programmes would need to define a priori the period of time to use for surveillance, and also to select a spatial scale at which to define hotspots. For varying time periods and spatial scales, we determined the % of excess malaria cases within the targeted hotspots compared with the surrounding area in the period of time immediately following the simulated surveillance.

One week periods of surveillance (top left panel of **Figure 4**) did not identify hotspots that are still present the following week at fine spatial scales (i.e., the plotted line indicates that the accuracy of targeting is 0% at scales of less than 1 km). On the other hand, at larger spatial scales we found that 1 week periods of surveillance were more accurate, resulting in the targeting of areas with a 60% excess of new malaria cases compared with the surrounding area at a scale of an 8 km diameter. A similar pattern was seen for monthly periods of surveillance. Longer surveillance periods (e.g., 6 months) resulted in targeting areas with an excess of 20% malaria cases compared with the surrounding area over the range of spatial scales examined.

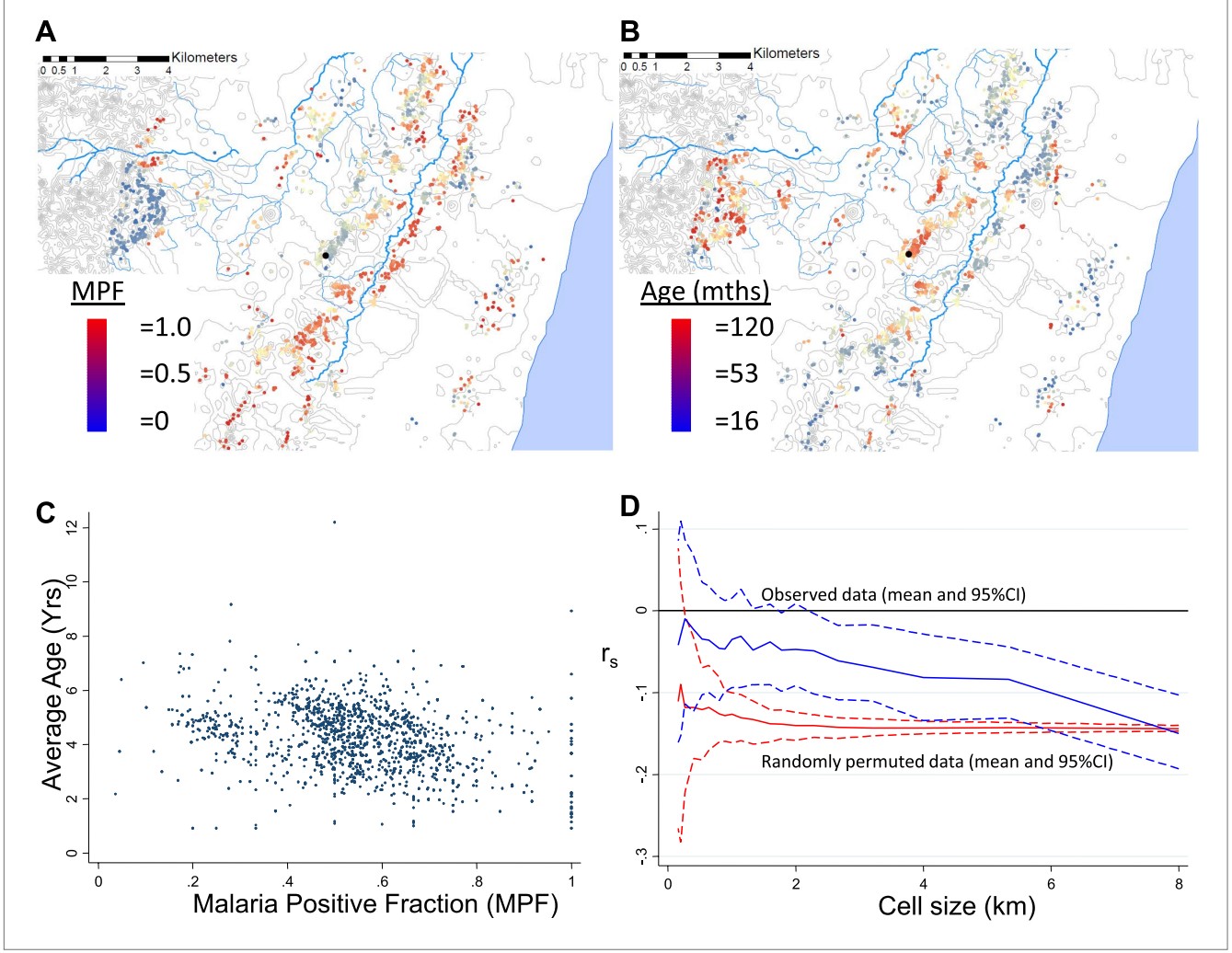

**Figure 1**. Geographical distribution of malaria positive fraction and average age of febrile malaria. Each plotted point represents an individual homestead, where the colour shading indicates the malaria positive fraction (MPF) in panel **A**, or the average age of children who test positive for malaria in panel **B**. Panel **C** shows the scatter plot for MPF vs average age (Spearman's rank correlation coefficient ($r_s$) = −0.16, p<0.0001). Panel **D** shows $r_s$ (y axis) plotted against scale of analysis (x axis), where a grid with varying cell size is imposed on the study area, $r_s$ is calculated within each cell and then the mean $r_s$ presented, with 95% confidence intervals produced by boot-strap (blue solid and dashed lines, respectively), and the results of analysis of spatially-random permutations of the data with equivalent cell size are shown for comparison (red solid and dashed lines, respectively). The analysis shown in panel **D** was compared on simulations with varying simulated characteristic scales, Signal:Noise ratios and with added gradients (*Figure 1— figure supplements 1–3*, respectively).

The following figure supplements are available for figure 1:

**Figure supplement 1**. Simulated data with varying imposed scales of clustering.

**Figure supplement 2**. Simulated data with varying signal to noise ratios.

**Figure supplement 3**. Simulated data with varying gradients around imposed scales of clustering.

## ITN use and spatial variation in risk

Mass distributions of Insecticide Treated Nets (ITNs) in the area began in 2006. ITN use was surveyed in 2009 and 2010. We found that children using ITNs had a reduced risk of malaria by logistic regression (i.e., OR = 0.69, 95%CI 0.67 to 0.8, p<0.001), in keeping with previous literature on the personal protection provided by ITN use (*Lim et al., 2011*). On the other hand, we did not identify significant

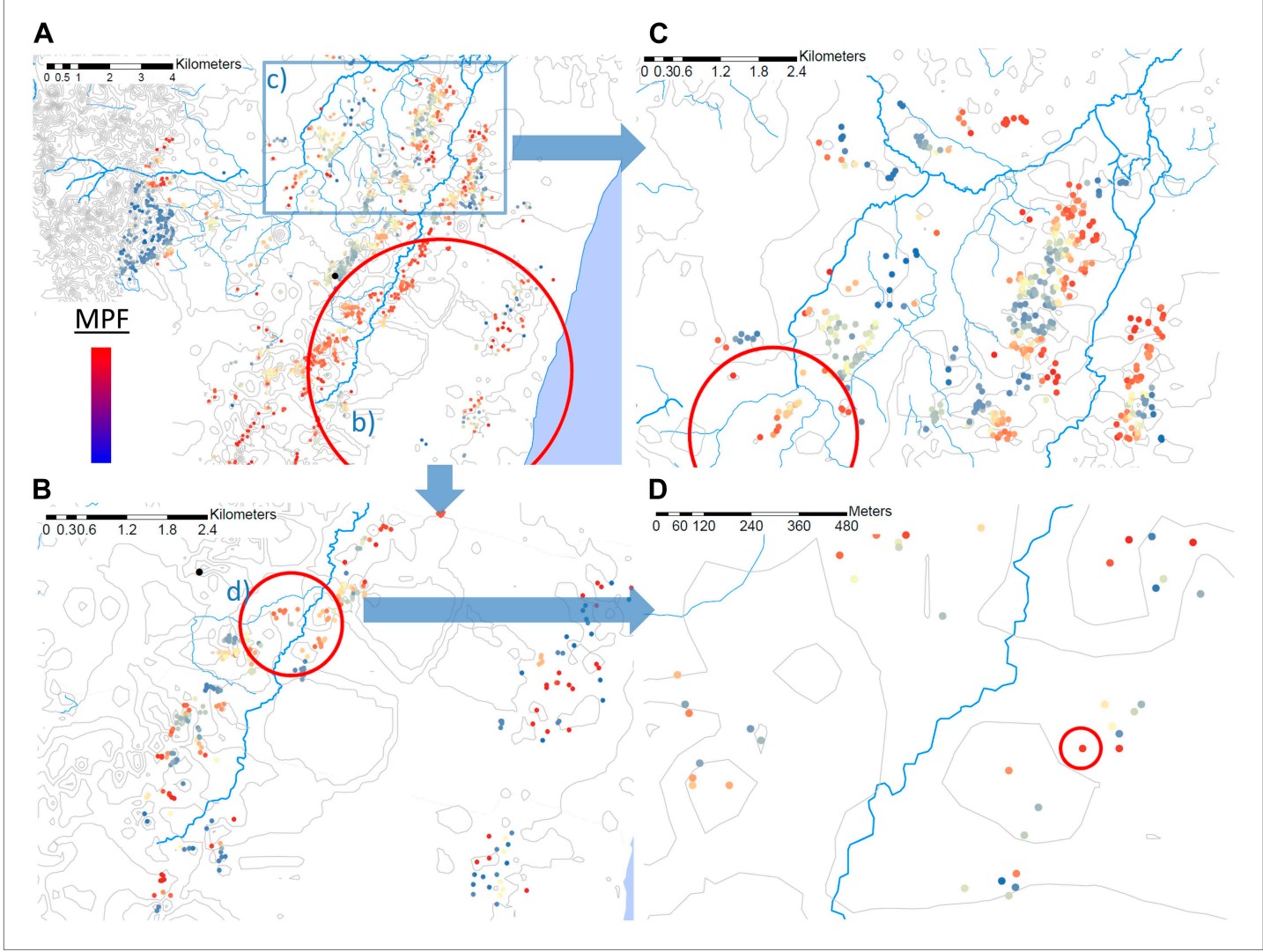

**Figure 2**. Hotspots within hotspots. Each plotted point represents an individual homestead, where the colour shading indicates the malaria positive fraction (MPF). Hotspots are identified using SATScan, using the whole study area (panel **A**), then repeated within the hotspot (panel **B**), within the hotspot of panel **B** (panel **D**), and then within a randomly chosen area outside the hotspot (panel **C**). The semi-variogram and log–log semi-variogram plot are shown in *Figure 2—figure supplements 1 and 2*, respectively.

The following figure supplements are available for figure 2:

**Figure supplement 1**. Semi-variogram.

**Figure supplement 2**. Log-log plot of semi-variogram.

evidence that ITN use was clustered spatially (Moran's I = 0.02, p=0.5). Furthermore, adding ITN use as a covariate in SaTScan analysis to locate hotspots had little effect on results; the addition of ITN use as a covariate changed the location of the hotspot by 120 m, and changed the predicted radius of the hotspot from 5.4 km to 5.2 km. On re-analysis of the homesteads within the 5.4 km hotspot, a further 0.87 km hotspot was identified the position and radius of which were not altered by the inclusion of ITN use as a covariate. Finally, within this 0.87 km hotspot the same 7 homesteads were identified as a hotspot irrespective of the inclusion of ITN use as a covariate. We did not identify significant evidence that ITN use correlated mean $MPF_{<1yr}$ ($r_s = -0.04$, p=0.04) or with the variance in $MPF_{<1yr}$ ($r_s = -0.01$, p=0.7). Hence, ITNs provided personal protection from malaria, but we were unable to show that they explained the spatial micro-epidemiological patterns.

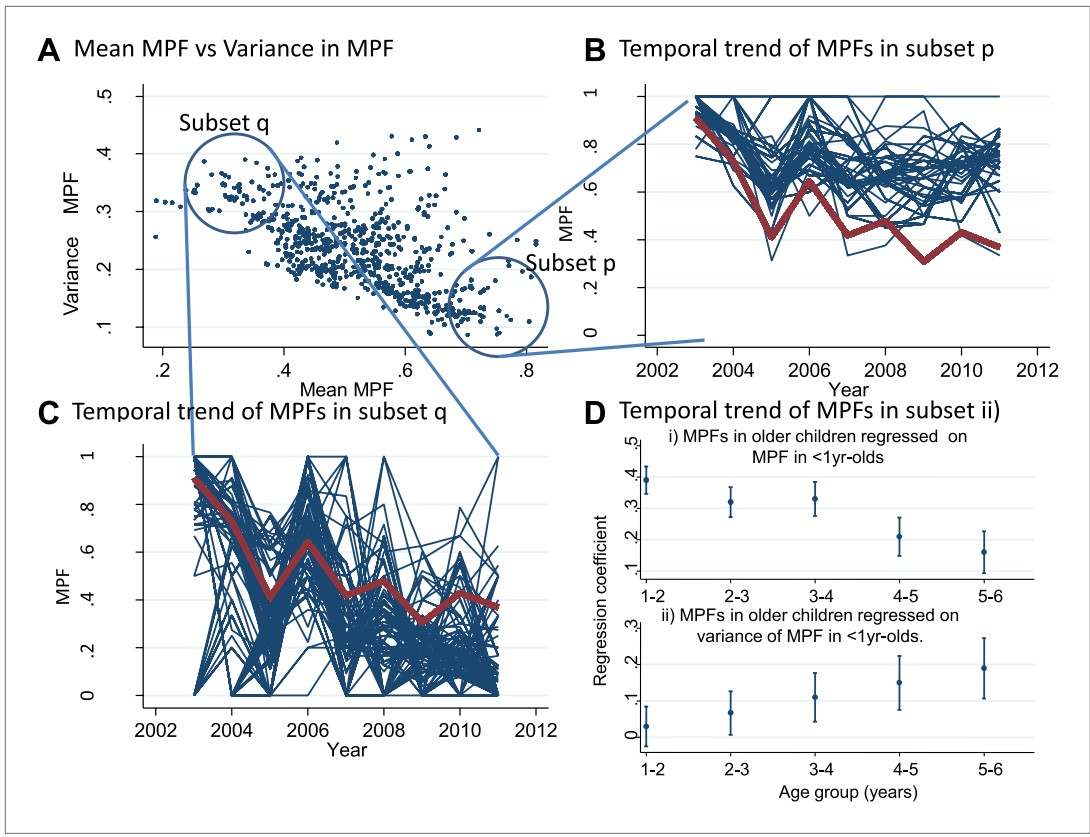

**Figure 3**. Temporal variations in malaria positive fraction. (Panel **A**) shows the scatter plot of individual homesteads by mean malaria positive fraction (MPF) on the x axis vs variance in MPF on the y axis ($r_s$ = −0.61, p<0.0001). A labelled blue circle indicates subset q (homesteads with high variance but low mean MPF) and subset p (homesteads with low variance and high mean MPF). The temporal trends for these two subsets are shown on panels (**B** and **C**), respectively. The median trend for the study area is shown in red. (Panel **D**) shows the regression coefficients (y axis) for the malaria positive fractions (MPF) in older children when regressed on; (i) the mean MPF in children <1 year of age ($MPF_{<1y}$) and (ii) MPF in older children when regressed on the variance in $MPF_{<1y}$ over the 9 years of the study. Separate multivariable regression models (i.e., with mean $MPF_{<1y}$ and variance in $MPF_{<1y}$ as explanatory variables) are fit for each age group as shown on the x axis (excluding children <1 year of age, whose data are used to calculate $MPF_{<1y}$).

## Discussion

We found that malaria cases were spatially heterogeneous in an 8-km radius area of coastal Kenya. The strongly significant inverse correlation between the malaria positive fraction (MPF) and average age of children presenting with malaria suggests variable acquisition of immunity between homesteads. Homesteads at high transmission intensity have a high MPF and a young average age of malaria (with older children becoming immune and therefore not presenting to the dispensary) whereas homesteads at low transmission intensity have a low MPF but an older average age of malaria since older children are not becoming immune as rapidly. In theory, this inverse correlation might have arisen because of heterogeneity at various spatial scales. For instance, there might have been a block of homesteads all at high transmission in one half of the study area (thus with high MPF and low average age) and a second block of homesteads at low transmission in the other half (with low MPF and high average age). On the other hand, the inverse correlation might have arisen because of a random distribution of 'high' and 'low' transmission intensity homesteads throughout the study area.

To determine at which spatial scale transmission was heterogeneous, we conducted an analysis where correlation coefficient was recalculated within each cell of a grid superimposed on the study area. The mean correlation coefficient of all cells was then presented as the cell size of the grid used was increased (*Figure 1D*). This analysis was done to identify the most influential geographical scale at

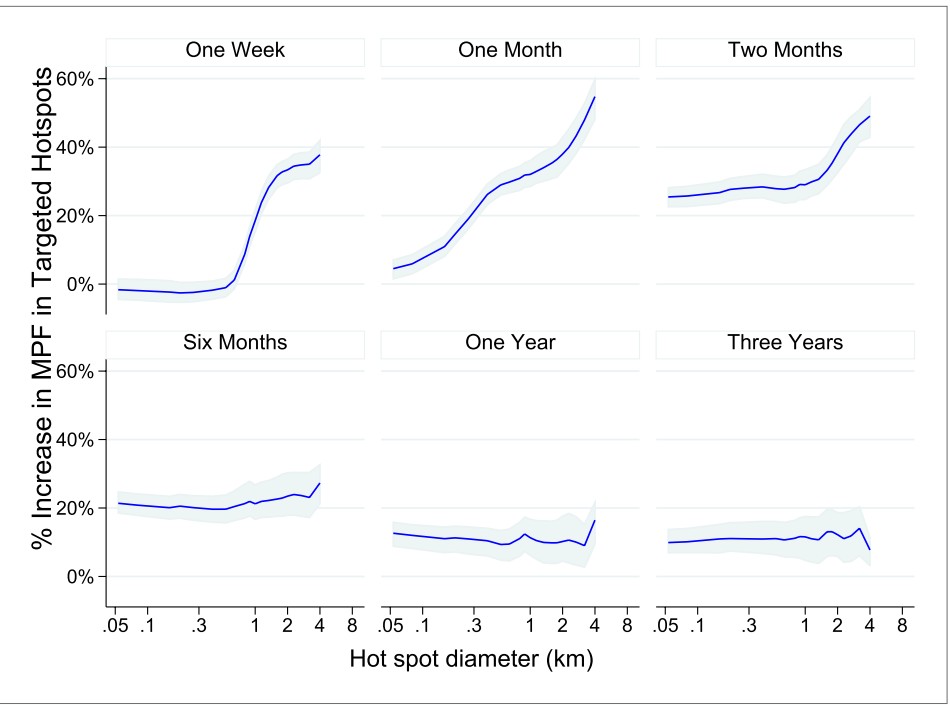

**Figure 4**. Theoretical accuracy of targeted control undertaken at varying temporal and spatial scales. The accuracy of varying strategies of hotspot identification is shown. Each panel is labelled with the time period of surveillance data used. The x axis shows the diameter of hotspot defined. In each case hotspots were selected to account for 20% of the homesteads in the area. The y axis shows the increase that would have been present assuming that they were targeted in the time period following their identification.

which the inverse correlation was observed. In simulated data, we noted 'spikes' where the inverse correlation was abruptly lost when the size of cells in the grid coincides with the size of the geographical 'blocks' of homesteads that drove the inverse correlation, as seen in *Figure 1—figure supplement 1*. Similar spikes were seen after adding simulated noise and gradients in space over which the correlation varied (*Figure 1—figure supplements 1, 2 and 3*). Real-world data would contain more complex sources of variation than we have simulated, and hence may not produce distinct spikes. Nevertheless, the analysis of these simulations suggests that discontinuities in the correlation between MPF and average age of malaria over cell size might be expected when clustering is at a specific spatial scale. In fact there was no such discontinuity in the function shown in *Figure 1D*, indicating that the inverse correlation was present at every geographical scale examined within our study. It is likely that this pattern would extend at greater geographical scales, since a similar inverse correlation between the age distributions of malaria cases and transmission intensity can be seen on comparing countries and regions (*Okiro et al., 2009*).

The pattern of spatial heterogeneity is relevant to malaria control, since targeted disease control is predicted to be highly effective (*Woolhouse et al., 1997*). Spatial targeting is particularly appropriate for malaria 'hotspots' (*Coleman et al., 2009*; *Moonen et al., 2010*; *Bousema et al., 2012*; *Sturrock et al., 2013*) and many malaria control programmes are already engaged in spatially-targeted intervention (*Zhou et al., 2010*; *Loha et al., 2012*). Our data showing clustering at varying spatial scales suggest that malaria control programs can expect to identify hotspots at many different geographical scales. We demonstrate that hotspots occur within hotspots, down to the level of a single homestead, and also that hotspots can be identified on 'zooming in' on random areas outside the main hotspot (*Figure 2C*). These hotspots were based on analysis of a large dataset with adequate power, and were strongly significant based on the multiple permutations run in SaTScan, suggesting that type I statistical error is an unlikely explanation for our findings. The complexity of presenting 'hotspots within hotspots' to a malaria control programme is further compounded by the temporal instability of the spatial pattern (*Figure 3*).

We therefore simulated the accuracy with which hotspots could be targeted using varying spatial scales and varying time periods of surveillance. We found that using data aggregated over 1 month of surveillance to define 4 to 8 km diameter hotspots would provide greatest accuracy, but this information is only relevant for 1 month before temporal instability necessitates further surveillance. One might therefore consider a continuous programme of parallel surveillance and targeting, where the surveillance data are examined at the end of each month to determine the location to be targeted for the following month. Continuous surveillance would allow adaptive targeting of hotspots for the following month. Such a strategy might be employed all year round, or for a limited period of the year depending on local seasonality. (*Cairns et al., 2012*) Targeting at this spatial scale has the added practical advantage that it could be done with village-level location data and would not require fine-scale geo-positional data.

There are some caveats to this recommendation. Our observations are from a single site. Other sites should examine their local data to determine whether a similar targeting strategy is appropriate. Furthermore, some hotspots did show temporal stability. For instance, we identified a 6 km diameter hotspot south east of the dispensary that maintained a 30–60% increase in MPF compared with the surrounding area throughout the 9-year surveillance.

Children with positive microscopy slides for malaria presenting at the dispensary may have genuine febrile malaria, or alternatively may have chronic asymptomatic parasitaemia with co-incident non-malarial fever. Previous studies estimating malaria attributable fractions in the locality suggest 61% of the children in our analysis would have malaria as the proximate cause of their illness, with the other 39% having chronic asymptomatic parasitaemia with co-incident fever from another cause (*Olotu et al., 2011*). We have previously demonstrated that spatial heterogeneity is more temporally stable when analysed for asymptomatic parasitaemia rather than febrile malaria (*Bejon et al., 2010*). Targeting hotspots of asymptomatic parasitaemia would require community surveys rather than dispensary monitoring, which may need to be done less frequently than monitoring of febrile malaria episodes.

Furthermore MPF is not a comprehensive indicator of transmission intensity. Homesteads with consistently low average ages of febrile malaria are likely to be stable high transmission homesteads (such as those in subset p of *Figure 3A*) which amplify transmission in the areas surrounding them. Targeting such high transmission homesteads to interrupt transmission may be highly effective (*Woolhouse et al., 1997*). The stronger inverse correlation between MPF and average age of febrile malaria as spatial scale increases (*Figure 1*) suggests that the spatial heterogeneity of transmission is progressively more stable at more coarse spatial scales.

Malaria transmission is determined by mosquito ecology and behavior. Mosquito ecology may be determined by obvious geographical features such as altitude (*Reyburn et al., 2005*), cultivation practices (*Lindsay et al., 1991*), streams and dams (*Ghebreyesus et al., 1999*), wind direction (*Midega et al., 2012*) and mosquito searching behaviour for hosts (*Smith et al., 2004*). Ecological models based on such features have been developed using frequentist techniques (*Omumbo et al., 2005*), Bayesian approaches (*Craig et al., 2007*), and fuzzy logic (*Snow et al., 1998*). However, the same ecological factor may act inconsistently in different geographical areas (*Kleinschmidt et al., 2001b*; *Gemperli et al., 2006*; *Noor et al., 2008*), and the effect of ecological factors is modified by fine-scale vector and host movement (*Perkins et al., 2013*). Our data suggests that the environmental factors determining malaria transmission operate at a range of spatial scales. We might speculate that mosquito breeding site density could be equally influenced by proximity to a large geographical feature such as a river, or to a micro-geographical feature such as a cow hoof-print (*Sattler et al., 2005*). Hence ecological models of malaria transmission will need to include data at a range of spatial scales in order to accurately predict malaria risk.

## Materials and methods

Approval for human participation in these cohorts was given by Kenya Medical Research Institute Ethics Research Committee, and research was conducted according to the principles of the declaration of Helsinki.

### Study population

Pingilikani Dispensary is 40 km to the North of Mombasa, in Kilifi Country, Coast Province, Kenya. The population relies mainly on subsistence farming and experiences all year round malaria transmission, with 'long' and 'short' rains each year causing two peaks in transmission. Estimates of the local EIR

were 22–53 in 2003 (1), and 21.7 infective bites per person per year in 2010 (2). Between 2003 and 2011, data were collected on all children (i.e., ≤15 years of age) attending the dispensary.

Demographic surveillance is conducted for the 240,000 people in a 900 square kilometre area in Kilifi County. Four-monthly enumeration rounds were conducted to identify births, deaths, and migration (3). Each inhabitant is described by their family relationships and their homestead of residence, with geospatial coordinates, and assigned a unique personal identifier. These details were used to link children visiting Pingilikani dispensary to geospatial coordinates for the homestead of residence. During enumeration rounds in 2009–2011 ITN use per individual was established during visits to the homestead, as reported by a homestead representative.

We restrict analysis to within an 8 km radius of the dispensary, which accounted for >96% of all visits to the dispensary and excluded visits with specific symptoms such as skin infections or cutaneous abscesses, otitis media, and gastroenteritis (>4 episodes diarrhoea per day) that might have been the primary motivation for seeking health care rather than fever per se. These latter exclusions combined accounted for 14% of all visits.

## Malaria diagnosis and treatment

All children presenting for assessment (except those with trauma as their only concern) had finger-prick blood samples examined for malaria parasites. Thick and thin blood smears were stained with 10% Giemsa and examined at x1000 magnification for asexual *Plasmodium falciparum* parasites. 100 fields were examined before slides could be considered negative. Amodiaquine was the first-line anti-malarial from 2003 to 2005, when policy changed to Co-artemether.

## Analysis

Fever was defined as either reported fever by the parents or measured fever, that is, axilliary temperature ≥37.5°C (*Mackowiak et al., 1997*). The malaria positive fraction (MPF) was calculated as the fraction of febrile children attending the dispensary with fever who were positive for malaria parasites by blood smear examination. MPF was aggregated by homestead. Multiple identifications of fever and parasitaemia in the same child within 21 days were considered a single episode.

The average age of febrile malaria was calculated as the arithmetic mean age at which children visited the dispensary with fever and malaria parasites. Correlations between average age of febrile malaria and MPF per homestead were calculated using spearman's rank correlation coefficient. Grids of gradually increasing cell size were calculated using longitude and latitude coordinates. Simulations were done using the distribution of homesteads identified in our study. We applied a factor to MPF (positive) and average age (negative) to the homesteads within a block of varying size to induce the appearance of clustering at a given spatial scale. Random noise was added to these simulations using a gamma distribution. In the first round of simulations we set the Signal:Noise ratio (i.e., the ratio between the factor applied to MPF and average age vs the mean amplitude of the noise) to reproduce the $r_s$ seen in the real data. In the second round of simulations, we varied the Signal:Noise Ratio as shown in individual panels, and in the third round of simulations we introduced a gradient over which the correlation emerged, where the factor applied to MPF and average age was tapered in a uniform way towards 1 beginning at the edge of the simulated block.

Hotspots were defined using SaTScan software to calculate the spatial scan statistic (*Kulldorff, 1997*). The software is freely available and can be downloaded from www.satscan.org. The version used in this analysis was downloaded in November 2012, as v9.1 for a 64-bit system. The spatial scan statistic uses a scanning window that moves across space. The scanning windows are circles centred on each homestead, with a radius varied from inclusion of only the single homestead it is centred on through to 30% of the population size. When using the Bernoulli model, the software calculates the fraction of cases/controls inside vs outside the each possible scanning window, and selects the window giving the highest probability of a case within the scanning window compared with the probability of a case outside the window. In our application of the Bernoulli model, cases were febrile children with parasitaemia and controls were febrile children without parasitaemia. The test of significance needs to take into account the whole process of selecting the optimal window rather than simply the comparison of inside vs outside the optimal window. This is achieved by running random permutations of the case/control data over the spatial co-ordinates of homesteads and determining the log-likelihood statistic for the model fit by the optimal window for each random permutation. The log-likelihood statistic for the real data is then compared with the statistics on

the random permutations to derive a p value. We used 9999 replications in our study. The maximum hotspot size was set at 30% of the population, and the inference level for significance was set at 0.05. The main analysis was done without adjustment for covariates, and a secondary analysis was conducted for the 2009/2010 data with and without ITN use as a covariate. Kernel smoothing with a 1 km radius is used for spatial display graphs, but all analyses of correlation are conducted on raw data without smoothing.

Semivariograms, Moran's I and linear regression models were run in Stata version 12 (StataCorp, Texas). Semivariograms were constructed using 0.1 km intervals between 0.1 km and 10 km. Moran's I was assessed globally using cumulative bands of <0.1, <0.5, <1 and <2 and <5 kms.

## Acknowledgements

Peter D Crompton is thanked for helpful comments during manuscript drafting. The manuscript is published with the permission of the Director of KEMRI. PB is jointly funded by the UK Medical Research Council (MRC) and the UK Department for International Development (DFID) under the MRC/DFID Concordat agreement. Work in Pingilikani was funded by the German Research Foundation (DFG, Grant number SFB 544, A7) and by the Wellcome Trust. Bonston Piri and Epson Mwadori are thanked for their contributions in making the geospatial data available. SIH is funded by a Senior Research Fellowship from the Wellcome Trust (095066).

## Additional information

### Funding

| Funder | Grant reference number | Author |
| --- | --- | --- |
| Wellcome Trust | 083579 | Philip Bejon |
| Medical Research Council (UK) | G1002624 | Philip Bejon |
| UK Department for International Development | G1002624 | Philip Bejon |
| German Research Foundation | SFB 544, A7 | Steffen Borrmann |

The funders had no role in study design, data collection and interpretation, or the decision to submit the work for publication.

### Author contributions

PB, PWG, BG, Conception and design, Analysis and interpretation of data, Drafting or revising the article; TNW, DB, MO, JP, MB, Acquisition of data, Drafting or revising the article; CN, Acquisition of data, Analysis and interpretation of data; SIH, DLS, Analysis and interpretation of data, Drafting or revising the article; TB, EB, SB, Conception and design, Acquisition of data, Drafting or revising the article; KM, Conception and design, Drafting or revising the article

### Ethics

Human subjects: Informed consent for participation was obtained, and specific ethical approval was obtained from the KEMRI Ethical Review Committee (SSC Protocol No. 2413: Spatial Epidemiology of Malaria Cases in the Kilifi District Demographic Surveillance Area). The KEMRI ethical review committee required that participants consent for participation in research and for their data to be stored, but does not require a further explicit statement consenting to publication. Our institutional guidelines would require this only in the event that individuals were identifiable in the publication.

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
