## [Decision Letter]

Thank you for sending your work entitled “The Fractal Micro-epidemiology of Malaria in Coastal Kenya” for consideration at *eLife*. Your article has been evaluated by a Senior editor, a Reviewing editor, and 2 reviewers. They agreed that the article presents interesting and potentially important findings. They have however a number of methodological concerns and questions regarding the patterns and their interpretation, which would need to be addressed before a decision can be reached.

The Reviewing editor and the other reviewers discussed their comments, and the Reviewing editor has assembled the following comments to help you prepare a revised submission. If you believe you can address these comments within a month, please submit your revision with a description of the changes.

The study draws on the sustained work over many years of an expert team who have documented the malaria parasitaemia status of children presenting to a clinic in rural Kenya. Finding hotpots of malaria transmission is not new, as the authors readily admit, but the phenomenon has rarely been as meticulously examined and analyzed as in this study.

1) The claim on the self-similarity of the patterns and the lack of a characteristic scale needs to be substantiated further for the analyses to be convincing. In particular, it is not clear that the results of the correlation analyses at different spatial resolutions could not arise from the combination of a characteristic scale and the increasing noise from fewer data as the resolution increases. The randomization test addresses the fewer homesteads for a random pattern but not for one with a characteristic scale. It would be valuable to impose a pattern with such a scale on the distribution of homesteads in the landscape and show that this does indeed create a 'shoulder'. Additional motivation and description of the methods is needed: how were the grids constructed? Why would randomly reassigning spatial co-ordinates to homesteads lead to a correlation that declines very rapidly with grid sizes, rather than remaining constant? The choice of analysis to demonstrate hotspots at every scale is unusual in the sense of involving a correlation between two quantities and how this changes with resolution. It is important that the expectations of different kinds of patterns be clear and that the significance of the observed pattern be established.

2) The title and Introduction emphasize the 'fractal' nature of the patterns. This term implies self-similarity in a stronger sense than the lack of an identifiable characteristic scale over some range of resolutions. In this sense, the calculation of a fractal dimension is important but has been relegated here to the Discussion with no actual evidence presented to demonstrate the dimension and the range of scales over which this number was obtained. This plot and the strength of the evidence for a fractal pattern need to be included.

3) The authors need to say where they obtained the SaTScan software, what version (or when downloaded). They also need a more detailed overview of how this method finds hot spots.

4) The possibility of conflation with mean age of infection should be considered. Mean age of infection results from the interaction between infection risk, reporting risk, and the mean age of children present. Age of children present could be an important confounder of the studied relationship: areas with younger children might have higher malaria rates and lower mean age of infection even if there is no signal from the direct causal relationship between them. Ideally, the authors would construct a statistic that measures age in the malarious population relative to the control population at each spatial scale. This issue must at least be carefully discussed.

5) The discussion of ITN use is not entirely convincing. Is there a way to include ITN use in the analysis, and thus really control for it? How much may the use of impregnated bednets be spatially or temporally heterogeneous? The correlation with variance in MPF seems like a red herring, and should be dropped.

6) It would be helpful to readers to know if, within the approximately 16 km diameter of the reported area, there are any obvious geographical differences – e.g., of altitude or ground water/foliage, that might affect malaria transmission. In the Discussion the authors state that 'Accurate characterization of these patterns will inform optimized surveillance and control policies'. Is it possible that such characterization within the study area would have produced results comparable with (and more easily obtained) than the data utilized from clinic records?

7) The investigators have data on the parasitaemia prevalence among children presenting without fever (and without trauma). The suggestion here is not that the authors repeat their analysis using parasitaemia prevalence in the non-febrile but that they discuss whether similar results might have been obtained. Alternatively, were parasite densities measured on blood films, so that the likelihood that a fever is actually due to malaria could be estimated?

8) The practical implications could be written more clearly. The principal suggestion is that “the use of one month cycles of surveillance to target 4 to 8 km diameter hotspots would be optimal” – some expansion on what is meant by “cycles” and “surveillance” - is it recommended that parasitaemia prevalences among febrile children should be the instrument of surveillance? And how frequently should the one-month surveys be carried out?

9) The authors need to be more careful throughout about “negative inference”. The null hypothesis is rarely true, and never confirmed. For example: the wording at the end of the second paragraph of the Introduction is far too strong – the authors should argue that this bias is likely to be small, not that it is unlikely to exist; similarly the “cannot be used” result is presumably a lack of statistical significance and does not imply there is no difference in subjective fever experience (it may imply that this difference is small, highly variable, or both); in the first paragraph of the Results, the authors should say MPF “was not found” to vary ; in the Results section “ITN use and spatial variation in risk”, the distribution was not “random” – the authors found no significant pattern, similarly they did not find a significant correlation with variance. The authors should be aware of this point when addressing the issue of non-malarial fevers, as well.

---

## [Author Response]

*1) The claim on the self-similarity of the patterns and the lack of a characteristic scale needs to be substantiated further for the analyses to be convincing. In particular, it is not clear that the results of the correlation analyses at different spatial resolutions could not arise from the combination of a characteristic scale and the increasing noise from fewer data as the resolution increases. The randomization test addresses the fewer homesteads for a random pattern but not for one with a characteristic scale. It would be valuable to impose a pattern with such a scale on the distribution of homesteads in the landscape and show that this does indeed create a 'shoulder'. Additional motivation and description of the methods is needed: how were*
*the grids constructed?*

We have done such simulations, using a variety of characteristic scales and signal:noise ratios, which are added to our revised manuscript as Supplementary Figures 1 and 2. We conducted these simulations by using the distributions of homesteads observed in the study area, and simulating an increase in malaria positive fraction (MPF) and decrease in age of children with febrile malaria with particular characteristic scale, and additional simulating a degree of random noise using a gamma distribution. We scaled the Signal:Noise ratio in each case to provide a Spearman’s rank correlation coefficient (r_s_) similar to that observed in our real data. We then analysed these simulated datasets in the same way as we had analysed our real data (i.e. examining the effect of spatial scale at which this correlation occurred by imposing increasingly fine-scale grids on the study area, calculating r_s_ within each cell of the grid, and then estimating the mean r_s_ at each scale of grid).

Interestingly, rather than a simple “shoulder” we see multiple “spikes” in r_s_ as the cell size of the grid varies. By examining the outcomes using different scales of simulated patterns, we found that a “spike” is observed where the scale of the simulated pattern coincides with the scale of the cell size of the superimposed grid (Figure 1—figure supplement 1). Further “spikes” then coincide with points at which the scale of the simulated pattern is a multiple of the grid size used.

We then investigated whether the ratio of signal to noise in the presence of a pattern at a characteristic scale might produce plots similar to those we observed in the real data. The results are shown in Figure 1—figure supplement 2. As expected, we observe that the correlation between age and the malaria positive fraction (i.e., r_s_) becomes more difficult to discern as the Signal:Noise ratio falls. However, spikes in r_s_ on varying the cell size of the grid can still be noted up until the point at which the Signal:Noise ratio completely obscures any correlation.

Furthermore, we investigated the effect of adding a gradient at the edge of the block of characteristic scale over which the clustering emerges (as a proportion of the block size). The results are shown in Figure 1—figure supplement 3. The presence of a gradient lowers the overall correlation seen, which has the effect of gradually attenuating the “spikes” but not of removing them entirely.

However, we recognise that simulations of noise and gradients are necessarily artificial and real-world data are likely to contain many other more complex sources of variation. We may expect less distinct peaks in such circumstances. Nevertheless these simulations all show marked discontinuities in the correlation between age of malaria and MPF over varying cell size of a superimposed grid. We suggest that a gradual emerging of the correlation between MPF and average age of malaria is not readily explained by clustering at a single spatial scale even in the presence of random variation or gradients.

Taking these findings together with the findings of a) hotspots at multiple spatial scales and b) the findings from the semi-variogram and log-log plot used to calculate a fractal dimension (see response below), we conclude that our findings are indeed the result of spatial clustering of transmission varying consistently at every scale examined.

We have therefore edited the text throughout to read “spikes” instead of “shoulders” and added text to the Results, Discussion, and Methods.

*Why would randomly reassigning spatial co-ordinates to homesteads lead to a correlation that declines very rapidly with grid sizes, rather*
*than remaining constant?*

Our reading of Figure 1) is that this is indeed what happens with the random simulation (shown in red). There is some apparent variation in r_s_ below a cell size of 0.5 km, but this is at a point where the confidence intervals of our estimate are quite wide (as a result of having few data points in each cell). It is the correlation in the observed data (shown in blue) that declines rapidly with reducing cell size.

*The choice of analysis to demonstrate hotspots at every scale is unusual in the sense of involving a correlation between two quantities and how this changes with resolution. It is important that the expectations of different kinds of patterns be clear and that the significance of the observed pattern be established*.

The results from simulations described above demonstrate the expectations with different kinds of pattern, and support the conclusion that that an inverse correlation between age and malaria positive fraction was present at every geographical scale examined within our study. This led us to predict that hotspots will occur at every scale, a conclusion that we empirically tested and confirmed as shown in Figure 2. This latter analysis does not depend on a correlation between two quantities but rather only considers the spatial clustering of febrile malaria cases. Similarly the analysis of the semivariogram, now added as Figure 2—figure supplement 1

(see point 2 below), suggests spatial clustering of febrile malaria cases at every spatial scale examined. Hence our conclusion of spatial clustering at every scale does not depend only on the inverse correlation between age and MPF, but on two further analyses (i.e., the demonstration of hotspots within hotspots in Figure 2, and the semivariogram shown in Figure 2—figure supplement 1), which do not involve a correlation between two variables.

*2) The title and Introduction emphasize the 'fractal' nature of the patterns. This term implies self-similarity in a stronger sense than the lack of an identifiable characteristic scale over some range of resolutions. In this sense, the calculation of a fractal dimension is important but has been relegated here to the Discussion with no actual evidence presented to demonstrate the dimension and the range of scales over which this number was obtained. This plot and the strength of the evidence for a fractal pattern need to be included*.

We have added the semivariogram plot and log-log semivariogram plot used to calculate the fractal dimension is shown in Figure 2—figure supplement 1 and Figure 1—figure supplement 2. The fractal dimension was calculated from the gradient of this line over the full range of the spatial scale displayed. We have added accompanying text to the manuscript in the Results and Methods.

*3) The authors need to say where they obtained the SaTScan software, what version (or when downloaded). They also need a more detailed overview of how this method finds hot spots*.

SaTScan software is free to download from the SaTScan website. Further details have been added to the relevant paragraph in the Methods section.

*4) The possibility of conflation with mean age of infection should be considered. Mean age of infection results from the interaction between infection risk, reporting risk, and the mean age of children present. Age of children present could be an important confounder of the studied relationship: areas with younger children might have higher malaria rates and lower mean age of infection even if there is no signal from the direct causal relationship between them. Ideally, the authors would construct a statistic that measures age in the malarious population relative to the control population at each spatial scale. This issue must at least be carefully discussed*.

The average age of children with non-malarial fever did not show any spatial clustering (Moran’s I=0.01, p=0.5 within 1 km and Moran’s I=0.02, p=0.5 within 5 km) and was not associated with MPF (r_s_=-0.02, p=0.4). We normalized age of febrile malaria for age of children with non-malarial fever by calculating the absolute difference between the two and found that there was still a negative correlation with MPF (r_s_=-0.08, p=0.011). However, the correlation is not as strong as that seen without normalization (r_s_) =-0.16, p<0.0001), partly because the normalization requires data in each homestead to calculate average age for children with and without malaria and therefore observations are dropped (i.e. ∼1,200 of ∼1,500 homesteads), and partly because normalizing according to the average age of non-malarial fever likely adds noise to the statistic. We therefore include the former results in the manuscript as reassurance that the studied relationship between MPF and average age of febrile malaria is unlikely to be confounded by spatial variation in average age of children in the community, but prefer to retain the use of average age of febrile malaria as our metric for the more detailed analysis according to spatial scale presented in the main body of the paper. We have added text to the Results section accordingly.

*5) The discussion of ITN use is not entirely convincing. Is there a way to include ITN use in the analysis, and thus really*
*control for it?*

ITN use can indeed be included as a covariate in the SaTScan analysis. Using the data from 2009 and 2010, we re-ran the hotspot analysis using SaTScan with and without the inclusion of ITN use as a covariate. The addition of ITN use as a covariate changed the location of the hotspot by 120m, and changed the predicted radius of the hotspot from 5.4 to 5.2km. On re-analysis of the homesteads within the 5.4km hotspot, a further 0.87km hotspot was identified the position and radius of which were not altered by the inclusion of ITN use as a covariate. Finally, within this 0.87km hotspot the same 7 homesteads were identified as a hotspot irrespective of the inclusion of ITN use as a covariate.

*How much may the use*
*of impregnated bednets be spatially or temporally heterogeneous?*

ITN use showed a random spatial distribution (Moran’s I=0.02, p=0.5). The Spearman’s rank correlation for proportion of homesteads using an ITN between these two years was 0.35 (p<0.0001) indicating that homestead use varied between years. Hence ITN use did not show an auto-correlation indicating non-random spatial heterogeneity, but was temporally heterogeneous. ITNs provided personal protection (i.e. OR=0.69, 95%CI 0.67 to 0.8, p<0.001), but adjusting for them did not appear to alter the location of hotspots (as described above).

*The correlation with variance in MPF seems like a red herring, and should be dropped*.

Indeed it was not our intention to imply that the correlation with variance in MPF was significant, and we have revised our wording to clarify this, and provide some context to our findings on personal protection of ITNs with regard to previous literature. Taking the three points above together, we have revised the paragraph on ITN use in the Results section.

*6) It would be helpful to readers to know if, within the approximately 16 km diameter of the reported area, there are any obvious geographical differences – e.g., of altitude or ground water/foliage, that might affect malaria transmission. In the Discussion the authors state that 'Accurate characterization of these patterns will inform optimized surveillance and control policies'. Is it possible that such characterization within the study area would have produced results comparable*
*with (and more easily obtained) than the data utilized from clinic records?*

There is only modest variation in altitude within the study area, from 30 to 180 metres above sea level (IQR 49-99 metres). Nevertheless, regarding the general point raised there is a previous literature that we now reference. Malaria transmission has been shown to vary by geographical features such as altitude (Reyburn et al), cultivation practices (Lindsay et al), streams and dams (Ghebreyesus et al). Ecological models have been developed using frequentist techniques (Omumbo et al), Bayesian approaches (Craig et al) and fuzzy logic (Snow et al). However, the same ecological factor may act inconsistently in different geographical areas (Gemperli et al, Noor et al, Kleinschmidt et al). Hence although there is obvious scope for analysis of our data in relation to geographical features, we believe this would be an appropriate topic for a further publication rather than an additional analysis to include here. On the other hand, there are some general points that we add to the discussion in response.

We identify a fractal pattern with clustering at multiple geographical scales. It therefore follows that in order to develop a model based on geographical features one would either need to identify a key feature that can be mapped on multiple geographical scales, or a combination of geographical features that can exist across multiple scales. For instance, we note a river in the Eastern part of the study area running North to South, and the main hotspot is associated with this river (Figure 2). However, the hotspot is associated with the Southern part but not the Northern part of the river, for reasons that are unclear to us. Furthermore, within the main hotspot there is further variation in risk (Figure 2) that does not appear to depend on distance from the river, but is located at a particular location along the river, with lower risk at other points along the river. Hence we reason that it is unlikely that any single geographical feature predicts a substantial portion of the variation in malaria risk, and any attempt to develop a geographical model needs to take into account the fractal pattern of spatial clustering. We summarize these considerations in a revised version of our final paragraph in the Discussion.

*7) The investigators have data on the parasitaemia prevalence among children presenting without fever (and without trauma). The suggestion here is not that the authors repeat their analysis using parasitaemia prevalence in the non-febrile but that they discuss whether similar results might have been obtained. Alternatively, were parasite densities measured on blood films, so that the likelihood that a fever is*
*actually due to malaria could be estimated?*

An analysis of asymptomatic parasitaemia would likely yield different results from an analysis of febrile parasitaemia, as we demonstrated in a previous study ([1] “Stable and unstable malaria hotspots in longitudinal cohort studies in Kenya”). However we do not have microscopy samples from an appropriate group in Pingilikani to limit analysis to asymptomatic parasitaemia. Samples from children attending with trauma have been taken as indicators of asymptomatic parasitaemia in the community, but as the reviewer notes samples from children with trauma were not examined. Children presenting without objective fever at the point of sampling in the dispensary are nevertheless presenting with an acute illness, and hence may not be very different from the objectively febrile group. We have shown in previous analysis of parasite densities from children presenting to the dispensary compared with parasite densities from cross-sectional surveys in the community that the malaria attributable fractions are substantial even among children without measured fever attending the dispensary (i.e. 40% (95%CI 39-41%), compared with 75% (95%CI 74-76%) Olotu et al 2011, “Defining Clinical Malaria: The Specificity and Incidence of Endpoints from Active and Passive Surveillance of Children in Rural Kenya”).

We do have parasite densities for the blood films here, but since we do not have appropriately matched community based sampling we cannot repeat a spatially explicit version of the malaria attributable fraction modelling in order to determine the likelihood that fever is actually due to malaria. Therefore we add to the Discussion accordingly.

*8) The practical implications could be written more clearly. The principal suggestion is that “the use of one month cycles of surveillance to target 4 to 8 km diameter hotspots would be optimal” – some expansion on what is meant by “cycles” and “surveillance” - is it recommended that parasitaemia prevalences among febrile children should be the instrument of surveillance? And how frequently should the one-month surveys*
*be carried out?*

We have re-written this section for clarity. We did indeed intend that monitoring parasitaemia among febrile children would be the instrument of surveillance. We intended that this would be done in cycles, with one month of monitoring, followed by one month of targeted intervention (during which parallel monitoring could take place in order to plan the following months targeting). These could in theory be undertaken throughout the year, or to conserve resources could be undertaken during the rainy season. We have revised the Discussion accordingly.

*9) The authors need to be more careful throughout about “negative inference”. The null hypothesis is rarely true, and never confirmed. For example: the wording at the end of the second paragraph of the Introduction is far too strong – the authors should argue that this bias is likely to be small, not that it is unlikely to exist; similarly the “cannot be used” result is presumably a lack of statistical significance and does not imply there is no difference in subjective fever experience (it may imply that this difference is small, highly variable, or both); in the first paragraph of the Results, the authors should say MPF “was not found” to vary ; in the Results section “ITN use and spatial variation in risk”, the distribution was not “random” – the authors found no significant pattern, similarly they did not find a significant correlation with variance. The authors should be aware of this point when addressing the issue of non-malarial fevers, as well*.

We accept this point and the necessary edits have been made throughout.